

# Quantifying the contribution of land use change to surface temperature in the lower reaches of Yangtze River

Xueqian Wang[1,2], Weidong Guo[1,2], Bo Qiu[1,2], Ye Liu[1,2], Jianning Sun[1,2], Aijun Ding[1,2]

1Institute for Climate and Global Change Research, School of Atmospheric Sciences, Nanjing University, Nanjing, China.
2Joint International Research Laboratory of Atmospheric and Earth System Sciences, Nanjing, China.

*Correspondence to:* Weidong Guo (guowd@nju.edu.cn)

**Abstract.** Anthropogenic land use has significant impact on climate change. Located in the typical East Asian monsoon region, the land-atmosphere interaction in the lower reaches of Yangtze River is even more complicated due to intensive human activities and different types of land use in this region. To better understand these effects on microclimate change, we compare differences in land surface temperature (Ts) for three land types around Nanjing from March to August, 2013, and then quantify the contribution of land surface parameters to these differences ($\Delta$Ts) by considering the effects of surface albedo, roughness length, and evaporation respectively. The atmospheric background contribution to $\Delta$Ts is also considered based on differences in air temperature ($\Delta$Ta). It is found that the cropland cooling effect and urban heat island effect both are induced by significant human activities in this region but they have opposite impacts on Ts. Various changes in surface parameters affect radiation and energy distribution and eventually modify Ts. It is the evaporative cooling effect that plays the most important role in this region. Besides, the background atmospheric circulation is also an indispensable part in land-atmosphere feedback induced by land use change and reinforces both cropland cooling and urban heat island effects.

## 1 Introduction

Land use/Land cover change (LULCC) has been widely investigated in the past few decades, and it has been found that more than half of the land surface on Earth has been exploited by human (Baldocchi, 2014). Robust evidences indicate that the impact of LULCC on temperature is obvious and this impact depends on different types of land surface transform. Deforestation usually has a warming effect at lower latitudes and a cooling effect at mid- to high latitudes (Lee et al., 2011). Global deforestation may result in cooling (Pitman et al., 2009;Davin and Noblet-Ducoudré, 2010;Betts et al., 2007) and amplify diurnal temperature variance (Alkama and Cescatti, 2016). The urban heat island (UHI) is one of the most significant human-induced phenomena and it usually results in apparent warming in urban area compared to the surrounding rural areas. The UHI effect depends on latitude, climate regime, urban area size, and time of the season (Kalnay and Cai, 2003;McCarthy et al., 2010;Zhao et al., 2014;Basara et al., 2008;Lin et al., 2016). Agriculture often leads to cooling temperature in different patterns, and the cooling effect can usually be magnified when it comes to irrigation (Campra et al., 2008;Kueppers et al., 2007;Lobell et al., 2006;Zhang et al., 2011). Thereby analyzing different types of land use plays an important role not only in




evaluating the climate change on different spatial scale (Alkama and Cescatti, 2016;Baldocchi and Ma, 2013;Huang et al., 2008;Wang et al., 2010;Hari et al., 2015), but also in improving the predictive capacity of models (Huang et al., 2015;Niu et al., 2011;Zhang et al., 2015). Although there have been many studies concentrating on LULCC, they rarely compare the differences in the mechanisms behind the land-atmosphere interaction with different types of land use.

The effects of anthropogenic land use on local climate are complicated with a series of stabilizing and reinforcing feedbacks (Baldocchi, 2014). Although the surface albedo change has been widely analyzed as the strongest climate forcing (Campra et al., 2008),IPCC (2013) emphasizes that it is not the only effect of LULCC because LULCC also causes other changes that don't affect the radiative process but can also significantly influence the surface temperature (Ts). These changes such as surface roughness (Davin and Noblet-Ducoudré, 2010;Kanda, 2007) and evapotranspiration changes (Pitman et al., 2009) are
more uncertain and difficult to quantify, whereas they exert essential influences on the radiative process and energy redistribution on the land surface (Baldocchi and Ma, 2013;Campra et al., 2008;Yang et al., 2014), and thereby cause obvious differences in Ts   over various land surface types under different climate backgrounds (Biggs et al., 2008;Luyssaert et al., 2014).

To understand the influence of LULCC, it is important to quantify the contributions of different surface parameters for each
type of land use. Juang (2007) proposed the method to decompose the observed change in Ts based on surface energy balance, and this method was refined later by Luyssaert et al. (2014). Lee (2011) presented a new metric and attributed the change in Ts to radiation, convection and evaporation. Chen and Dirmeyer (2016) added the atmospheric background effect to the metric proposed by Lee. This method can be used to calculate each factor's contribution to Ts in areas with different vegetation cover (Bright et al., 2014;Li et al., 2015) as well as urban area (Zhao et al., 2014).

The lower reaches of Yangtze River Valley, which is located in the typical East Asian monsoon region, is one of the regions with the most intensive human activities around the world. Rapid urbanization, industrialization, expansion of farmland, animal husbandry, deforestation and afforestation are common features in this region. In monsoon region, LULCC affects climate not only by influencing local convection through radiation and surface heat fluxes, but also by influencing the monsoon onset and weakening and related precipitation (Hsu and Liu, 2003;Xue et al., 2004). However, both flux observations
and characteristic analyses are very limited in the lower reaches of Yangtze River Valley, let alone quantitative analysis (Gao, 2003;Bi et al., 2007). In this study, the contributions of different surface land parameters to surface temperature are calculated based on analysis of data collected at several sites, where the land use type includes crop, grass and urban area respectively (Guo et al., 2016). We first quantitatively compare the influences of several different surface parameters on Ts over different types of managed land, and then demonstrate that the Bowen ratio effect dominates the feedback of land use change to surface
temperature in this region, while other factors play a secondary role.



## 2. Data and methods

### 2.1 Observation Sites and data

The measurements used in this study were collected at three sites in the lower reaches of Yangtze River. The urban site, where the average building height is 19.7m, is located at Dangxiao, the central urban area of Nanjing (32°2′24″N,118°47′24″E). The other two sites are both located at around (31°43′08″N,118°58′51″E ) in Lishui county and classified as a grassland site and a cropland site, respectively. The grass height is about 60cm. Rice grows in the summer (mid June to early November) and wheat grows in the winter (from mid- to late November to early June of next year) nearby the cropland site, with the largest plant height of 75cm.

In this study, sensible and latent heat fluxes are measured at 30-min intervals by the eddy covariance system (EC3000, Campbell) deployed at 3 m height over the grass site and crop site, and at 36.5 m height above the 22 m high building at the urban site. The sampling frequency is 10Hz for measurements by the Data acquisition (CR5000). Strict correction and quality control have been applied to all the flux measurements (Foken et al., 2004). The measurements contain micro-meteorological elements of air temperature (HMP45C-L, Vaisala), precipitation (TE525MM-L, Texas Electronics), and surface radiation fluxes including downward and upward short-wave (CM21, Kipp & Zonen) and long-wave (CG4, Kipp & Zonen) fluxes at half-hour intervals. Additional information about these observations can be found in the previous studies (Guo et al., 2016). The analysis focuses on March to August in 2013. This is because the eddy covariance method is assumed to work well only when turbulence can fully develop. To quantify the different contributions to ΔTs more accurately, we use Integrated Turbulence Characteristics (ITC) proposed by Foken (Foken and Wichura, 1996) to select data for general use (ITC<100%). Such standard was also adopted by FLUXNET program (Foken et al., 2004).

### 2.2 Methodology

In an ideal state, the surface energy balance can be expressed as:

$$R_n + AH = H + LE + G \qquad (1)$$

Where Rn is the net radiation calculated from $R_n = DSR + DLR - USR - ULR$,DSR, DLR, USR and ULR are the downward shortwave radiation, downward longwave radiation, upward shortwave radiation and upward longwave radiation, respectively. Anthropogenic heat (AH) flux is more obvious in urban areas than in rural areas but it is difficult to accurately measure. H and LE are the sensible and latent heat flux. G includes the heat flux at the surface of soil or buildings and the thermal storage in the canopy and it's relatively small. In this paper, we only discuss the differences between Rn, LE and H on the basis of the observations at the urban area of Nanjing and the countryside.

Following the method proposed by Lee (2011) and refined by Chen and Dirmeyer (2016), the biophysical mechanism can be expressed as a temperature change and decomposed into three direct factors, i.e. radiation balance, aerodynamic resistance and





evaporation, and one indirect factor of air temperature on larger scale. Therefore, ignoring AH and G in urban area, it can be approximated by:

$$\Delta T_s \approx \frac{\lambda_0}{1+f}\Delta S + \frac{-\lambda_0}{(1+f)^2}R_n^*\Delta f_1 + \frac{-\lambda_0}{(1+f)^2}R_n^*\Delta f_2 + \Delta T_a \quad (2)$$

with

$$f = \frac{\lambda_0 \rho C_p}{r_a}(1+\frac{1}{\beta})$$

$$\Delta f_1 = \frac{-\lambda_0 \rho C_p}{r_a}(1+\frac{1}{\beta})\frac{\Delta r_a}{r_a}$$

$$\Delta f_2 = \frac{-\lambda_0 \rho C_p}{r_a}\frac{\Delta \beta}{\beta^2}$$

Where $\Delta T_s$ is the difference in the surface temperature between other managed sites and natural grass site, $\lambda_0 = 1/4\varepsilon\sigma T^3$ is the local climate sensitivity, $f$ is the energy redistribution factor, $\Delta S = DSR - USR$ is net shortwave radiation, $R_n^* = (1-\alpha)DSR + DLR - (1-\varepsilon)DLR - \varepsilon\sigma T_a^4$ is the parent net radiation, $\alpha = USR/DSR$ is the albedo, $\varepsilon$ is the surface emissivity, $\sigma$ is the Stefan-Boltzmann constant. $T_a$ is the air temperature at reference height and $\Delta T_a$ is the

10 difference between managed sites and grass site.

We regard the grass site, with local native vegetation, as the base site. The terms on the right-hand side of Eq. (2) shows that the contributions to $\Delta T_s$ are from radiation change (term 1), aerodynamic resistance change (term 2) related to aerodynamic resistance ($r_a$) which represents the surface roughness effect, and evaporation change (term 3) related to Bowen ratio ($\beta = H/LE$). Therefore, all the independent parameters of the land use type and the respective contribution of them to $T_s$

15 can be calculated.

In the sites covered by vegetation, the aerodynamic resistance can be expressed as (Verhoef and De Bruin, 1997):

$$r_a = \frac{1}{\kappa u_*}[\ln\frac{z_m - d}{z_{0m}} + \ln\frac{z_m}{z_{0h}} - \Psi_h(\zeta)] \quad (3)$$

Where $Z_{0m}$ is the aerodynamic roughness length, which can be given by the independent method (Chen et al., 1993); $\Psi_h(\zeta)$ is the stability correction function for temperature; and $\ln\frac{z_{0m}}{z_{0h}} = 0.13(\frac{z_{0m}u_*}{v})^{0.45}$ (Zeng and Dickinson, 1998),

20 where $v$ is the viscosity coefficient with a value of 1.46×10-5m2s-1 . But in urban area, because the wind profile is not applicable well, we calculate the aerodynamic resistance from:





$$r_a = \frac{\rho C_p (T_s - T_a)}{H} \quad (4)$$

## 3. Results

### 3.1 Differences in surface temperature

Due to the East Asian monsoon anomaly and decreased moisture convergent, 2013 is an extremely drought year in southern China, where the summer precipitation decreased by more than 78% of the average amount and broke the historical record over the past 50 years (Yuan et al., 2016). The drought in 2013 was especially severe in the mid- to lower reaches of Yangtze River. Under the same dry condition, different land use types cause different feedbacks to surface temperature (Ts) and other surface characteristics. To compare the influence of different land use type on microclimate, the surface temperature change ( $\Delta T_s$ ) from grassland to cropland and to urban area are quantified.

Monthly variations of Ts differences ($\Delta T_s$) between crop and grass sites and between urban and grass sites are presented in Figure 1. During the entire growing season, cropland had an obvious cooling effect, which was strengthened when it came to irrigation(Kueppers et al., 2007;Lobell et al., 2006). The extremely large differences between crop and grass sites were -1.75℃ in April and -2.46℃ in August (Figure 1a) with less precipitation in these months (Guo et al., 2016). However, the cooling effect of only -0.34℃ in June was relatively small because wheat harvest and straw burning increased Ts in the cropland site. On the contrary, the urban heat island (UHI) effect resulted in at least 1℃ higher temperature at the urban site than at the rural sites in each month of the growing season. The extremely warm and dry condition in April and July was more evident in urban area than at the grassland site, with the maximum value of 1.95℃ higher temperature in April and 2.17℃ in July. Comparing different land types, it is clear that land use influences the local Ts to a large extent and makes it more complicated. Cropland cooling and UHI effects are both obvious in East Asian monsoon region.

### 3.2 Variations and differences in land surface factors

The characteristics of physical processes at different surface types can be represented by surface parameters, including albedo, Bowen ratio, surface roughness and aerodynamic resistance. These parameters reflect the momentum, heat and moisture exchanges between land and atmosphere (Baldocchi and Ma, 2013;Bright et al., 2015;IPCC, 2013). Figure 2 shows the monthly variation and differences of these parameters across the crop, urban and grass sites. Except for the extremely low albedo in cropland from May to June, the differences in albedo, Bowen ratio and surface roughness between crop site and grass site are opposite to the differences between urban site and grass site.

Monthly variation of surface albedo shows that the albedo in grassland gradually decreased from March to June but slightly increased in July and August because of the drought. Due to a series of agricultural activities including wheat harvest, straw burning and rice irrigation from early May to mid June, the albedo at cropland decreased quickly and then increased when rice





started growing. Thereby the difference in albedo ($\Delta\alpha$) between the crop and grass site was negative from May to July, with the extreme value of -0.06 in June. Monthly $\Delta\alpha$ between urban and grass site remained negative during the whole growing season (Figure 2b). As a measurement of dry and wet condition of the surface to a certain degree, Bowen ratio decreases when there is sufficient soil water content. The largest differences occurred in March, with a value of 2.8 at the urban site and -1.24 at the

crop site. With the lack of precipitation in August, the increase in $\beta$ obviously occurred at the grassland site but not at the other two managed land sites (Figure 2c). The Bowen ratio at the crop site was always low in the growing season because of sufficient water supply.

Besides, Figure 2e and 2f present that the urban surface roughness ($Z_{0m}$) is much higher than that at the lands with vegetation cover. The average surface roughness length at the urban area is 2.82m higher than at the suburban area. When it comes to the

sites with vegetation cover, it is shown that $Z_{0m}$ at the grassland site was a little higher than that at the cropland site and the extreme difference was -0.05m in June due to the wheat harvest. Contrary to the differences in $Z_{0m}$, the aerodynamic resistance at the urban site was obviously lower than that at other sites during the entire growing season. The grass site and crop site had a similar trend of aerodynamic resistance in the spring but a relatively large difference in the summer. Different to the $Z_{0m}$ variation, the aerodynamic resistance in grassland was much higher than that in urban area but a little lower than that in

cropland. The largest differences in aerodynamic resistance between grassland and urban area and that between grassland and cropland both occurred in August with values of -44.36 s/m and 29.08 s/m respectively.

### 3.3 Attribution of the differences in micrometeorological elements

In the land-atmosphere interaction process under the same climate background, different types of land use with different surface parameters can affect the radiation budget and redistribution of surface sensible and latent heat flux, and eventually

affect local surface temperature. Figure 3 shows the attribution of $\Delta Ts$ to both direct surface parameters and indirect atmospheric effect at the crop and urban sites. The $\Delta Ts$ attributed to roughness was calculated by aerodynamic resistance. Thus negative value means high roughness and cooling effect. It is clear that the dominant modification was caused by the evaporation represented by Bowen ratio, the value of which was even comparable to the observed $\Delta Ts$ in the lower reaches of Yangtze River. While the $\Delta Ts$ driven by surface roughness and evaporation were of opposite sign at the crop site and the urban

site, contributions of the two parameters are both strengthened from the spring to summer. Even though the low vegetation height with low $Z_{0m}$ at the crop site was favorable for higher $\Delta Ts$, evaporation based on sufficient water supply reduced the Bowen ratio and cooled Ts efficiently in the summer.

Averages of observed Ts in the growing season were -1.79℃ at the crop site and 2.01℃ at the urban site. At the crop site, the calculated Ts was -1.76℃, albedo and aerodynamic resistance contributions were 0.09℃ and 0.47℃, respectively, but Bowen

ratio cooling effect decreased $\Delta Ts$ by -1.40℃. At the urban site, the calculated Ts was 1.25℃ and the difference between the observed and calculated values, which was larger in the summer, was partly derived from the ignorance of heat storage and anthropogenic heating. Even if radiation and surface roughness cooling existed, the limited evaporation reduced the partitioning of Rn to latent turbulent heat flux and warmed the urban area by 2.29℃.



Atmospheric feedback is also important. It not only can change the cloud distribution due to water and heat differences or aerosol effects and impact solar radiation (Yang et al., 2012;Betts et al., 2007;Biggs et al., 2008), but also can affect circulations or the variation of vegetation physical properties such as albedo and evaporation (Niu et al., 2011;Yang et al., 2014) and subsequently affect Ts. The atmospheric background effects of Ta were relatively stable and could not be neglected during
the whole growing season. It had an average contribution of -0.93℃ to the cropland cooling effect and 0.54℃ to the urban heat island effect respectively and enlarged the difference in surface temperature induced by land use.

**4 Conclusions and Discussions**

Our study presented the first-handed observational evidences to verify the model results. Located in East Asian monsoon region, the lower reaches of Yangtze River has experienced the most intensive land use changes around the world, which has
significant impacts on the local and regional climate. However, these impacts may not be easy to quantify due to the lack of observations in this region and uncertainties in modeling results. We used in-situ data to quantify the contributions of two main land use types here, the irrigated cropland and the rapid urbanization, to the microclimate change. It shows that the crop cooling and UHI were both obvious..The differences in Ts were larger in the months with low precipitation and the monthly maximum values at both sites are even larger than 2℃.

For the study of LULCC effects on regional climate, more attention should be paid to nonradiative forces and the feedbacks from the background circulation. Although the surface albedo change caused by LULCC has been considered to be the strongest climate forcing and its effect has been widely and quantitatively estimated, other non-radiative modifications induced by LULCC including the roughness and evaporation are also important. Our results shows that the alteration of radiation, aerodynamic resistance, evaporation and air temperature all contributed to ΔTs (Figure 3). Despite the negative contributions
of net solar radiation and aerodynamic resistance, the positive contribution of Bowen ratio controlled both the cropland cooling effect and urban heat island effect which have been enlarged by the influence of background atmospheric circulation.

These results clearly demonstrate that evaporative cooling effect is the most important factor that modifies the surface temperature change in the lower reaches of Yangtze River valley, and the temperature change induced by this effect is even comparative to the total value of ΔTs. Recent studies (Chen and Dirmeyer, 2016;Zhao et al., 2014) indicate that surface
roughness usually dominates the land-atmosphere feedback of deforestation and urbanization in North America. Although the evaporative cooling and surface roughness both are important in land-atmosphere interaction, even more than albedo changes in some regions, their effects usually cannot be revealed accurately by models (IPCC, 2013) and the studies of these surface parameters effects are still insufficient, especially in some regions with scarce in-situ observations such as in the lower reaches of Yangtze River. To better understand the local and regional climate change and the possible large scale feedback, for
example the feedback between land use change and the East Asian monsoon system, more observational data and accurate modeling studies of the physical mechanisms between the land surface and the atmosphere are needed for further theoretical analysis.



## Acknowledgments

This research is jointly sponsored by Natural Science Foundation of China (Grant No. 41475063, 91544231), the National Science and Technology Support Program (2014BAC22B04). This work is also supported by the Jiangsu Collaborative Innovation Center for Climate Change. For data used in our study, please contact the corresponding author: Weidong Guo

(guowd@nju.edu.cn).

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





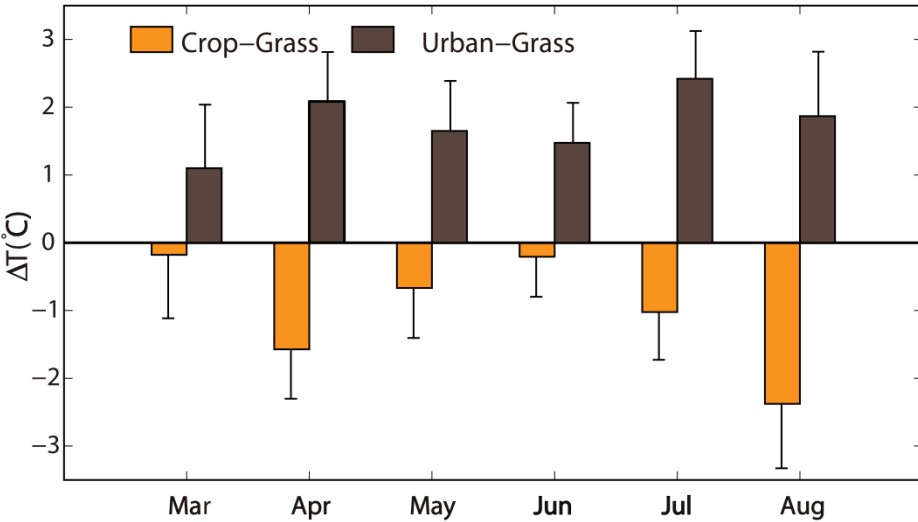

Figure 1: Differences in surface temperature between different sites in Nanjing from March to August 2013.

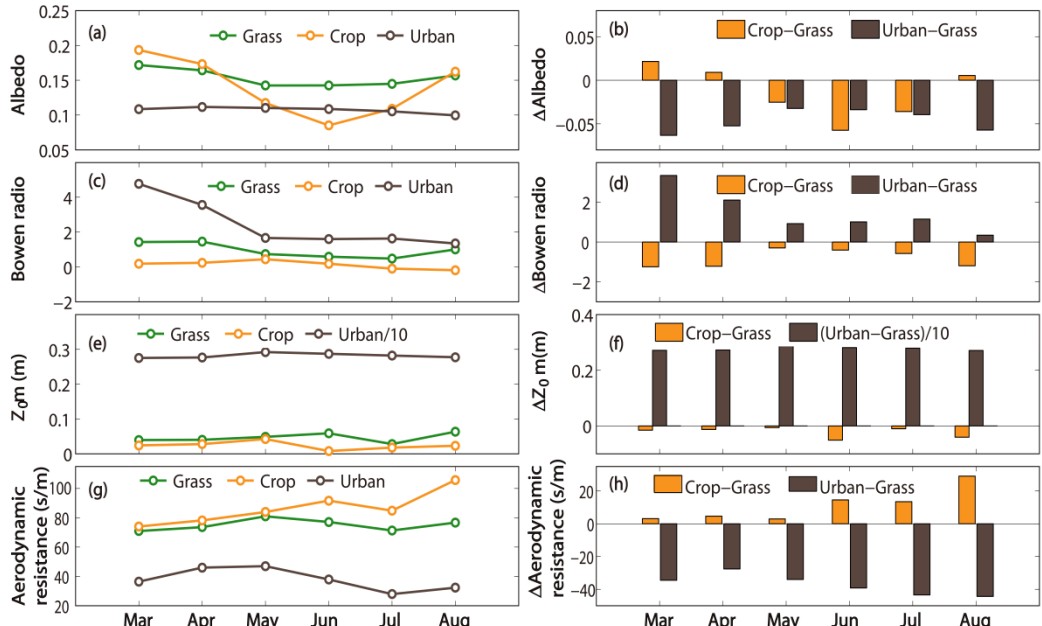

**Figure 2: Monthly variations of different factors at the three sites and the differences between the other two sites and the grass site in Nanjing from March to August 2013: (a,b) albedo, (c,d) Bowen ratio, (e,f) surface roughness, and (g,h) aerodynamic resistance.**





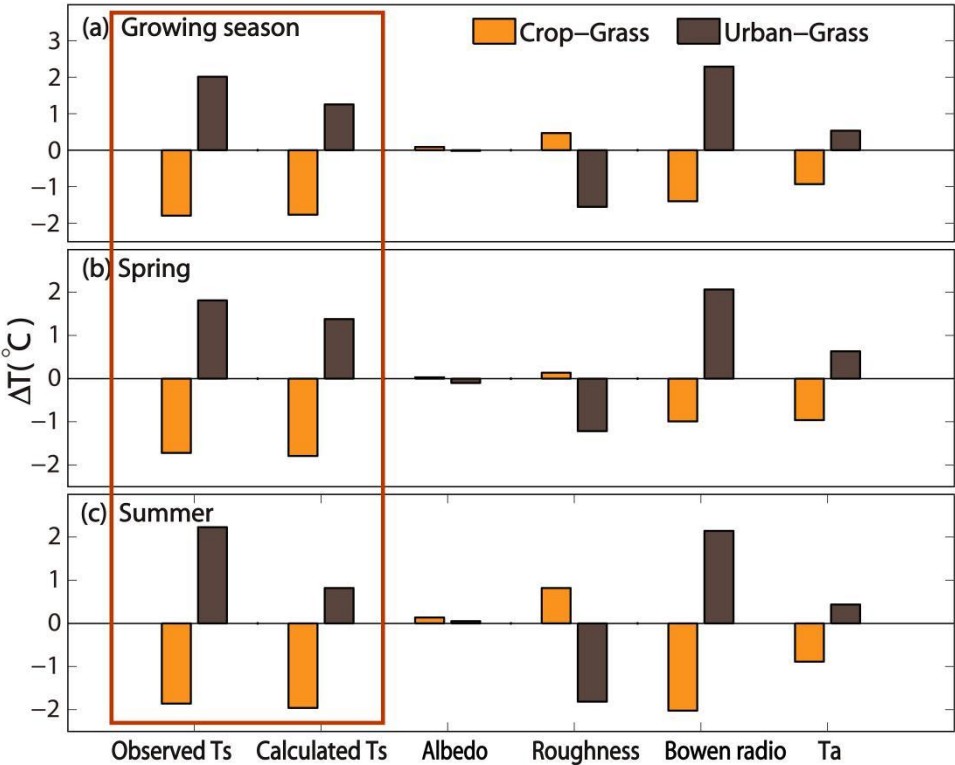

Figure 3: Contributions to the differences in surface temperature between urban and cropland sites and the grassland site due to radiation, aerodynamic resistance , evaporation, and air temperature (Ta) in (a) growing season, (b) spring and (c) summer, 2013.