# Peer review of "Quantifying the contribution of land use change to surface temperature in the lower reaches of Yangtze River"

_Atmospheric Chemistry and Physics, 2016_

## Referee Comment (RC1) · Anonymous Referee #1 · 23 Dec 2016

General comment:

Land use/cover change (LUCC) has been known as one of the most important anthropogenic forcings to local and regional climate change. LUCC, resulting in the changes of land surface parameters such as albedo, Bowen ratio, surface roughness, and aerodynamic resistance, could induce the variations of surface heat fluxes, and ultimately change land surface temperature. Most previous studies of LUCC focused on modeling land surface processes, but the uncertainties in climate models limit our understanding of its impacts. However, few researches have used the field measurements for different types of land use to investigate the biophysical mechanisms behind LUCC.

Using in-situ data from three sites with different land cover types and the attribution

method proposed by Lee (2011), this study quantified the contributions of different land surface parameters and background climate to the changes in land surface temperature over cropland, grassland and urban area. This paper is well organized, with some interesting results. However, some of the analyses seem not so solid and need further revisions. I would recommend it to be published with minor revisions as specified below.

Specific comments:

(1) It would be helpful if more quantitative information can be provided in the "Abstract".

(2) It would be meaningful to show the site locations and the spatial distribution of land covers over the lower reaches of Yangtze River.

(3) Page 2, Line 24: should be "weakening related precipitation".

(4) Figure 1: what do the lines of error bar indicate? How to calculate the uncertainty of temperature change?

(5) As shown in Figure 2, the albedo in grassland is different from that in cropland, especially in June. This could lead to different energy distributions in these two sites, eventually the changes in surface temperature. However, Figure 3 shows that the contribution of surface albedo to the temperature changes over cropland is the least .Why?

(6) This study suggested that the effect of evaporation cooling dominates the change in surface temperature. However, some studies based on Lee's method reported different findings in other regions. It is worth comparing the results of this study with previous ones.

(7) Eq (2): $\Delta S$ used in the calculation is the difference of net shortwave radiation between managed site and grass site. Is this mismatched?

(8) Page 5, Line 16: the authors only show the differences in surface temperature. No

more information on other atmospheric variables (e.g., humidity, precipitation) is given. It could not indicate that "extremely warm and dry condition in April and July was more evident in urban area than at the grassland site".

(9) Page 5, Line 4: It isn't reasonable to state that "Bowen ratio decrease when there is sufficient soil water content", since other factors also can change Bowen ratio.

(10) Page 6, Line 15: The sentence "The largest differences in aerodynamic resistance between grassland and urban area and that between grassland and cropland..." seems to be contradictory to Figure 2.

(11) Page 6, Line 28: "observed Ts" and "calculated Ts" should be "observed $\Delta$Ts" and "calculated $\Delta$Ts". Figure 3 shows the differences in Ts, not the original values.

(12) It would be helpful to change Ts and Ta to Ts and Ta.

(13) Please change Lee or Lee (2011) to Lee et al. (2011).

(14) Figure 3: It would be better if the scale ranges of y-axis were the same.

---

## Referee Comment (RC2) · Anonymous Referee #2 · 8 Feb 2017

General comments: As one of the most developed regions in China, the middle to lower reaches of Yangtze River is featured by intense human activities, especially large scale urbanization and agricultural processes in terms of land use and land cover change (LUCC). It goes without saying that such alternations will inevitably exert distinct influences on the exchange of water and energy fluxes between land surface and near surface atmosphere at local to regional scales. Therefore, it has significant scientific value to access the contribution of above mentioned influences quantitatively. At the same time, it is still a very challenging issue. A method proposed by Lee (2011) was introduced in the manuscript to quantitatively quantify the contribution of land surface parameters (albedo, aerodynamic roughness length, and Bowen Ratio) to the changes

of surface temperature associated with LUCC under the similar background of microm-eteorology based on field observations from 3 sites at lower reaches of Yangtze River. This work is important to further understand the effects of LUCC on regional climate over typical regions undergoing rapid economic developments.

Specific Comments: Some questions that need to be further addressed are listed as follows: 1. To my understanding, terms in Eq. (2) are originally at the interval of every half hour, then averaged as the monthly mean. Please add more details to let us know how you select the data, especially the land surface turbulence data. Some descriptions on the details of monthly mean value in Fig.2 are also needed;

2. P4L9, "parent", should be "apparent";

3. P6L28,29,30, "Ts" at X axis of Fig.3 should be delta_Ts;

4. Is it suitable to regard "Bowen ratio" as a land surface parameter? Or it is more likely to be regarded as a kind of land surface characteristics? Considering the fact that changes of evaporation in cropland are mainly affected by human activities like irriga-tion, Bowen Ratio may reflect some characteristics of underlying surface rather than a single parameter. Attentions should be paid to this issue throughout the analyses section of the paper.
* * *

---

## Author Comment (AC1) · 15 Mar 2017

Response: We would like to thank the referee for providing the insightful suggestions, which indeed help us reconsider and further explore the underlying problems in quantifying the relative contributions to local surface temperature change in the lower reaches of Yangtze River. In the revised manuscript, we have added more clear descriptions on the physical characteristics, in-depth discussion of different factors' effects on surface temperature change, and compared the results with previous studies. The revisions corresponding to each specific comment are tracked in the marked-up manuscript respectively.

Specific comments: (1)It would be helpful if more quantitative information can be pro-

vided in the "Abstract". ResponseïiǰAccepted. We have added more quantitative information in the section of Abstract.

(2)It would be meaningful to show the site locations and the spatial distribution of land covers over the lower reaches of Yangtze River. ResponseïiǰYes. The locations and spatial distribution of the sites and land covers have been shown on the other paper ( Guo et al., 2016) which can be find in our manuscript. We restated it in P4L20. Reference Guo, W., Wang, X., Sun, J., Ding, A., and Zou, J.: Comparison of land-atmosphere interaction at different surface types in the mid- to lower Yangzi River Valley, Atmospheric Chemistry & Physics, 16, 9875-9890, 10.5194/acp-2016-49, 2016, 2016.

(3)Page 2, Line 24: should be "weakening related precipitation". ResponseïiǰAccepted.

(4)Figure 1: what do the lines of error bar indicate? How to calculate the uncertainty of temperature change? ResponseïiǰThe Error bars in Fig.1 represent 1 s.d. of the daily surface temperature change for each month. We have made an explanation in P8L3.

(5)As shown in Figure 2, the albedo in grassland is different from that in cropland, especially in June. This could lead to different energy distributions in these two sites, eventually the changes in surface temperature. However, Figure 3 shows that the contribution of surface albedo to the temperature changes over cropland is the least. Why? ResponseïiǰIt is true that there is a large difference in albedo between cropland and grassland, especially in June. However, net radiation, the source of the energy distribution, is not just related to the shortwave radiation, but also the longwave radiation. And other surface characteristics also effect energy distribution. The surface temperature changes are more sensitive to the evaporation and surface roughness than albedo in the lower reaches of Yangtze River. Even the difference in albedo is large, its contribution is small. This is also the case for the differences in albedo between urban area and grassland.

(6)This study suggested that the effect of evaporation cooling dominates the change in surface temperature. However, some studies based on Lee's method reported different findings in other regions. It is worth comparing the results of this study with previous ones. ResponseïijŽYes. We have added more descriptions about previous studies in other regions and compared them with ours in the section of Conclusions and Discussions.

(7)Eq (2): S used in the calculation is the difference of net shortwave radiation between managed site and grass site. Is this mismatched? ResponseïijŽWe have corrected it.

(8) Page 5, Line 16: the authors only show the differences in surface temperature. No more information on other atmospheric variables (e.g., humidity, precipitation) is given. It could not indicate that "extremely warm and dry condition in April and July was more evident in urban area than at the grassland site". ResponseïijŽThe difference of humidity between grass and urban area has been shown in our previous study (Guo et al., 2016). We have added the reference for this sentenceãĂĆ Reference Guo, W., Wang, X., Sun, J., Ding, A., and Zou, J.: Comparison of land-atmosphere interaction at different surface types in the mid- to lower Yangzi River Valley, Atmospheric Chemistry & Physics, 16, 9875-9890, 10.5194/acp-2016-49, 2016, 2016.

(9) Page 5, Line 4: It isn't reasonable to state that "Bowen ratio decrease when there is sufficient soil water content", since other factors also can change Bowen ratio. ResponseïijŽWe have rephrased this sentence as "Sufficient soil water content can benefit the energy exchange in the way of higher LE and lower Bowen ratio".

(10) Page 6, Line 15: The sentence "The largest differences in aerodynamic resistance between grassland and urban area and that between grassland and cropland..." seems to be contradictory to Figure 2. ResponseïijŽIt has been rewritten as " ...between urban area and grassland, and that between cropland and grassland..." .

(11) Page 6, Line 28: "observed Ts" and "calculated Ts" should be "observed Ts" and "calculated Ts". Figure 3 shows the differences in Ts, not the original values.

ResponseïijŽWe have replaced the "observed Ts" and "calculated Ts" as "observed $\Delta$Ts" and "calculated $\Delta$Ts"

(12) It would be helpful to change Ts and Ta to $\Delta$Ts and $\Delta$Ta. ResponseïijŽIt has been changed in our revised manuscript.

(13) Please change Lee or Lee (2011) to Lee et al. (2011). ResponseïijŽAccepted. We have replaced it.

(14) Figure 3: It would be better if the scale ranges of y-axis were the same. ResponseïijŽThanks. We have used the same scale ranges in Fig. 3.

Please also note the supplement to this comment:
http://www.atmos-chem-phys-discuss.net/acp-2016-1013/acp-2016-1013-AC1-supplement.pdf

**Supplement:**

[revised manuscript text omitted]

---

## Author Response (AR1)

**Response to Referee #1**

*Land use/cover change (LUCC) has been known as one of the most important anthropogenic forcings to local and regional climate change. LUCC, resulting in the changes of land surface parameters such as albedo, Bowen ratio, surface roughness, and aerodynamic resistance, could induce the variations of surface heat fluxes, and ultimately change land surface temperature. Most previous studies of LUCC focused on modeling land surface processes, but the uncertainties in climate models limit our understanding of its impacts. However, few researches have used the field measurements for different types of land use to investigate the biophysical mechanisms behind LUCC. Using in-situ data from three sites with different land cover types and the attribution method proposed by Lee (2011), this study quantified the contributions of different land surface parameters and background climate to the changes in land surface temperature over cropland, grassland and urban area. This paper is well organized, with some interesting results. However, some of the analyses seem not so solid and need further revisions. I would recommend it to be published with minor revisions as specified below.*

**Response:** We would like to thank the referee for providing the insightful suggestions, which indeed help us reconsider and further explore the underlying problems in quantifying the relative contributions to local surface temperature change in the lower reaches of Yangtze River. In the revised manuscript, we have added more clear descriptions on the physical characteristics, in-depth discussion of different factors' effects on surface temperature change, and compared the results with previous studies.

*Specific comments:*

*(1) It would be helpful if more quantitative information can be provided in the "Abstract".*

**Response:** Accepted. We have added more quantitative information in the section of Abstract.

*(2) It would be meaningful to show the site locations and the spatial distribution of land covers over the lower reaches of Yangtze River.*

**Response:** Yes. The locations and spatial distribution of the sites and land covers have been shown on the other paper ( Guo et al., 2016) which can be find in our manuscript. We restated it in P4L20.

Reference

Guo, W., Wang, X., Sun, J., Ding, A., and Zou, J.: Comparison of land-atmosphere interaction at different surface types in the mid- to lower Yangzi River Valley, Atmospheric Chemistry & Physics, 16, 9875-9890, 10.5194/acp-2016-49, 2016, 2016.

*(3) Page 2, Line 24: should be "weakening related precipitation".*

**Response:** Accepted.

*(4) Figure 1: what do the lines of error bar indicate? How to calculate the uncertainty of temperature*

*change?*

**Response:** The Error bars in Fig.1 represent 1 s.d. of the daily surface temperature change for each
month. We have made an explanation in P8L3.

*(5) As shown in Figure 2, the albedo in grassland is different from that in cropland, especially in June.*
*This could lead to different energy distributions in these two sites, eventually the changes in surface*
*temperature. However, Figure 3 shows that the contribution of surface albedo to the temperature*
*changes over cropland is the least. Why?*

**Response:** It is true that there is a large difference in albedo between cropland and grassland, especially
in June. However, net radiation, the source of the energy distribution, is not just related to the shortwave
radiation, but also the longwave radiation. And other surface characteristics also effect energy
distribution. The surface temperature changes are more sensitive to the evaporation and surface
roughness than albedo in the lower reaches of Yangtze River. Even the difference in albedo is large, its
contribution is small. This is also the case for the differences in albedo between urban area and
grassland.

*(6) This study suggested that the effect of evaporation cooling dominates the change in surface*
*temperature. However, some studies based on Lee's method reported different findings in other regions.*
*It is worth comparing the results of this study with previous ones.*

**Response:** Yes. We have added more descriptions about previous studies in other regions and compared
them with ours in the section of Conclusions and Discussions.

*(7) Eq (2): S used in the calculation is the difference of net shortwave radiation between managed site*
*and grass site. Is this mismatched?*

**Response:** We have corrected it.

*(8) Page 5, Line 16: the authors only show the differences in surface temperature. No more information*
*on other atmospheric variables (e.g., humidity, precipitation) is given. It could not indicate that*
*"extremely warm and dry condition in April and July was more evident in urban area than at the*
*grassland site".*

**Response:** The difference of humidity between grass and urban area has been shown in our previous
study (Guo et al., 2016). We have added the reference for this sentence。

Reference

Guo, W., Wang, X., Sun, J., Ding, A., and Zou, J.: Comparison of land-atmosphere interaction at
different surface types in the mid- to lower Yangzi River Valley, Atmospheric Chemistry & Physics, 16,
9875-9890, 10.5194/acp-2016-49, 2016, 2016.

*(9) Page 5, Line 4: It isn't reasonable to state that "Bowen ratio decrease when there is sufficient soil*
*water content", since other factors also can change Bowen ratio.*

**Response:** We have rephrased this sentence as "Sufficient soil water content can benefit the energy exchange in the way of higher LE and lower Bowen ratio".

*(10) Page 6, Line 15: The sentence "The largest differences in aerodynamic resistance between grassland and urban area and that between grassland and cropland..." seems to be contradictory to Figure 2.*

**Response:** It has been rewritten as " ...between urban area and grassland, and that between cropland and grassland..." **.**

*(11) Page 6, Line 28: "observed Ts" and "calculated Ts" should be "observed Ts" and "calculated Ts". Figure 3 shows the differences in Ts, not the original values.*

**Response:** We have replaced the "observed Ts" and "calculated Ts" as "observed $\Delta T_s$" and "calculated $\Delta T_s$"

*(12) It would be helpful to change Ts and Ta to $T_s$ and $T_a$.*

**Response:** It has been changed in our revised manuscript.

*(13) Please change Lee or Lee (2011) to Lee et al. (2011).*

**Response:** Accepted. We have replaced it.

*(14) Figure 3: It would be better if the scale ranges of y-axis were the same.*

**Response:** Thanks. We have used the same scale ranges in Fig. 3.

**Response to Referee #2**

*As one of the most developed regions in China, the middle to lower reaches of Yangtze River is featured by intense human activities, especially large scale urbanization and agricultural processes in terms of land use and land cover change (LUCC). It goes without saying that such alternations will inevitably exert distinct influences on the exchange of water and energy fluxes between land surface and near surface atmosphere at local to regional scales. Therefore, it has significant scientific value to access the contribution of above mentioned influences quantitatively. At the same time, it is still a very challenging issue. A method proposed by Lee (2011) was introduced in the manuscript to quantitatively quantify the contribution of land surface parameters (albedo, aerodynamic roughness length, and Bowen Ratio) to the changes of surface temperature associated with LUCC under the similar background of micrometeorology based on field observations from 3 sites at lower reaches of Yangtze River. This work is important to further understand the effects of LUCC on regional climate over typical regions undergoing rapid economic developments.*

**Response:** We would like to thank the referee for providing the insightful suggestions, which indeed help us reconsider and further explore the underlying problems in quantifying the relative contributions to local surface temperature change in the lower reaches of Yangtze River. In the revised manuscript, we have added more clear descriptions on data selection and mechanism analysis to better understand the effects of different land conversions on climate change. The revisions corresponding to each specific comment are tracked in the revised manuscript respectively.

*Specific Comments:*

*Some questions that need to be further addressed are listed as follows:*

*1. To my understanding, terms in Eq. (2) are originally at the interval of every half hour, then averaged as the monthly mean. Please add more details to let us know how you select the data, especially the land surface turbulence data. Some descriptions on the details of monthly mean value in Fig.2 are also needed;*

**Response:** We used the selected data at half-hour intervals to obtain the daily averages of albedo, Bowen ratio and aerodynamic resistance. Then we calculated their contributions to $\Delta T_s$ based on Eq. (2) and made a discussion on monthly scale.

Quality assessment/quality control of land surface turbulent data is crucial for the reliability of results achieved from observations of surface fluxes. In our manuscript, we adopted the ITC approach proposed by Foken et al. (2004) to select well-developed land surface turbulent data. We added the details of data selection and calculation in the section of Data and Method.

We averaged the daily values to discuss the monthly variations of different factors. Different land surface types have different surface color, permeable rate, heat content and surface roughness, which results in the different properties and impacts in the land-atmosphere interaction. Human modifications in the urban area make it more obviously different from that in grassland and cropland, especially in the surface roughness. Some human activities also make the special case in the cropland, like the extremely low albedo in June due to the straw burning. More details of monthly mean value in Fig.2 has been discussed in the part of section 3.2.

*2. P4L9, "parent", should be "apparent";*

**Response:** It has been corrected in P6L1.

*3. P6L28,29,30, "Ts" at X axis of Fig.3 should be delta_Ts;*

**Response:** Thanks. We have replaced it P9, Line18-21.

*4. Is it suitable to regard "Bowen ratio" as a land surface parameter? Or it is more likely to be regarded as a kind of land surface characteristics? Considering the fact that changes of evaporation in cropland are mainly affected by human activities like irrigation, Bowen Ratio may reflect some characteristics of underlying surface rather than a single parameter. Attentions should be paid to this issue throughout the analyses section of the paper.*

**Response:** Accepted. This study attributes surface temperature change to different factors. Albedo represents the effect of radiative forcing, Bowen ratio and aerodynamic roughness represent the effect of energy distribution together. It is true that Bowen ratio is not just a single parameter. It reflects the characteristics both associated with the water content and temperature difference between land and atmosphere. We have changed it as a "factor" that effects on $\Delta T_s$ in our revised manuscript.

[revised manuscript text omitted]